# Benefits and harms of perioperative high fraction inspired oxygen for surgical site infection prevention: a protocol for a systematic review and meta-analysis of individual patient data of randomised controlled trials

Stijn W de Jonge [1,2,3] Rick H Hulskes [1,2] Maedeh Zokaei Nikoo,[4] Robert P Weenink,[2] Christian S Meyhoff,[5] Kate Leslie,[6] Paul Myles [7] Andrew Forbes,[8] Robert Greif,[9] Ozan Akca [10] Andrea Kurz,[4,11] Daniel I Sessler,[4,11] Janet Martin,[12] Marcel GW Dijkgraaf [13,14] Kane Pryor [15] F Javier Belda,[16,17] Carlos Ferrando,[18,19] Gabriel M Gurman,[20] Christina M Scifres,[21] David S McKenna,[22] Matthew TV Chan,[23] Pascal Thibon,[24] Jannicke Mellin-Olsen,[25] Benedetta Allegranzi,[26] Marja Boermeester,[1,3] Markus W Hollmann[2]

**Correspondence to**
Dr Stijn W de Jonge;
s.w.dejonge@amsterdamumc.nl

## ABSTRACT

**Introduction** The use of high fraction of inspired oxygen (FiO$_2$) intraoperatively for the prevention of surgical site infection (SSI) remains controversial. Promising results of early randomised controlled trials (RCT) have been replicated with varying success and subsequent meta-analysis are equivocal. Recent advancements in perioperative care, including the increased use of laparoscopic surgery and pneumoperitoneum and shifts in fluid and temperature management, can affect peripheral oxygen delivery and may explain the inconsistency in reproducibility. However, the published data provides insufficient detail on the participant level to test these hypotheses. The purpose of this individual participant data meta-analysis is to assess the described benefits and harms of intraoperative high FiO$_2$ compared with regular (0.21–0.40) FiO$_2$ and its potential effect modifiers.

**Methods and analysis** Two reviewers will search medical databases and online trial registries, including MEDLINE, Embase, CENTRAL, CINAHL, ClinicalTrials.gov and WHO regional databases, for randomised and quasi-RCT comparing the effect of intraoperative high FiO$_2$ (0.60–1.00) to regular FiO$_2$ (0.21–0.40) on SSI within 90 days after surgery in adult patients. Secondary outcome will be all-cause mortality within the longest available follow-up. Investigators of the identified trials will be invited to collaborate. Data will be analysed with the one-step approach using the generalised linear mixed model framework and the statistical model appropriate for the type of outcome being analysed (logistic and cox regression, respectively), with a random treatment effect term to account for the clustering of patients within studies. The bias will be assessed using the Cochrane risk-of-bias tool for randomised trials V.2 and the certainty of evidence using Grading of Recommendations, Assessment,

## STRENGTHS AND LIMITATIONS OF THIS STUDY

⇒ Individual participant data meta-analysis (IPD MA) of (quasi-)randomised controlled trials provides the best possible analysis of the available data on the participant level, permitting the investigation of potential effect modifiers.
⇒ IPD MA requires the collaboration of all investigators that have published data on the relevant topic and leads to a broad consensus on the outcome and interpretation of the analysis
⇒ IPD MA depends on the quality of data that is made available by the authors of the original studies.

Development and Evaluation methodology. Prespecified subgroup analyses include use of mechanical ventilation, nitrous oxide, preoperative antibiotic prophylaxis, temperature (<35°C), fluid supplementation (<15 mL/kg/hour) and procedure duration (>2.5 hour).

**Ethics and dissemination** Ethics approval is not required. Investigators will deidentify individual participant data before it is shared. The results will be submitted to a peer-review journal.

**PROSPERO registration number** CRD42018090261.

## INTRODUCTION

Surgical site infection (SSI) is one of the most common healthcare-associated infections and leads to morbidity, mortality and longer hospital stay.[1–4] The attributable costs can be more than €14 000 per SSI, and European totals are estimated to range from €1.5 to €19 billion per year.[5 6] In 2016–2017,

both the WHO and the Centers for Disease Control and Prevention (CDC) independently released evidence-based guidelines on the prevention of SSI that included a recommendation in favour of the administration of high fraction of inspired ($FiO_2$) for patients undergoing surgery under general anaesthesia.[7–9] This has led to a debate between opponents and proponents of the use of high $FiO_2$ in several editorials and correspondences across medical specialty literature.[10–20] Concerns were raised on the safety of the use of high $FiO_2$ as well as on the conflicting study results with some in support of the use of high $FiO_2$ to reduce SSI and some not.[10–20] Finally, studies by of one of the authors that contributed to the body of evidence were retracted because of unreproducible statistics.

In response to these concerns, the WHO conducted an independent systematic review on the safety of high intraoperative $FiO_2$ and updated the systematic review on its effectiveness, excluding the disputed trials.[21 22] No evidence of harm to discourage the use of high $FiO_2$ was found, yet the evidence of an effect of SSI had become weaker, and the recommendation was adjusted accordingly.[23] Despite various studies and recommendations, there is still no consensus on the safety and effectiveness of using high $FiO_2$ during surgery with regard to SSI, all-cause mortality and other adverse events in adult patients. This leads to practice variation that inevitably exposes patients to suboptimal care.[24] There is a need for better understanding and consensus on this issue.

Since the early promising results, perioperative care has changed considerably. Open abdominal surgery has been largely replaced by laparoscopic surgery, fluid management has moved from liberal to restrictive, to advanced goal directed regimens and active perioperative warming has become a mainstay.[25–27] All these changes have considerable consequences for haemodynamic, microcirculation and eventually peripheral oxygen delivery.[28–30] These changes may explain the inconsistency in reproducibility, but the available data provides insufficient detail on the participant level to test the potential of high $FiO_2$. Meta-analysis of individual participant data uses the raw individual-level data from the original study for synthesis and overcomes this limitation.[31 32] Individual participant data meta-analysis (IPD MA) enables analysis of uniform outcomes with more statistical power and assessment of potential effect modifiers.[33 34] Importantly, IPD MA requires collaboration with all published researchers on the topic leading to a broad consensus on the outcome of data analysis and interpretation.

The purpose of this IPD MA is to assess the potential benefits and harms of intraoperative high (0.60–1.00) $FiO_2$ compared with traditional (0.21–0.40) $FiO_2$ and its effect modifiers in adult patients undergoing surgery with SSI being the primary outcome. This IPD MA is initiated by the University of Amsterdam/Amsterdam University Medical Center, and encouraged by the WHO and the World Federation of Societies of Anaesthesiologists (WFSA) to provide patients and practitioners with

the best possible evidence and guidance on this disputed area and will give clearance of the disputed hypothesis that high $FiO_2$ reduces the incidence of SSI.

## METHODS AND ANALYSIS
### Protocol and registration
This study protocol was registered with the International Prospective Register of Systematic Reviews (PROSPERO) on 7 March 2018 and was last updated on 15 July 2022 (registration number CRD42018090261). The study protocol is designed and written to adhere to the Preferred Reporting Items for Systematic Review and Meta-Analysis (PRISMA) Protocols[35] and the PRISMA of individual patient data (IPD).[32]

### Patient and public involvement statement
This project is encouraged by the WHO and the WFSA to provide patients and practitioners with the best possible evidence and guidance on this disputed area. WHO and WFSA have provided external independent review and advice on research direction and aim.

### Governance
This study is an initiative of the Amsterdam University Medical Centre, encouraged by the WHO and the WFSA. Both organisations recognise the urgent need for this research and provide external independent review and advice. The writing committee consists of the study coordinator, two reviewers, a lead methodologist and a principal investigator from both the surgery and the anaesthesiology department of the Amsterdam University Medical Centre and two external content matter experts. The writing committee is entirely independent of the initial trials and has full responsibility for all methodological decisions. A broader steering committee with representatives of the collaborating trial groups identified during the project will be invited to comment on and coauthor the final protocol and IPD MA report. By sharing their IPD, collaborators will obtain one coauthorship on the IPD MA report and one additional coauthorship if data of more than 300 participants is shared. For transparency and against intellectual bias, a record will be kept of all comments. Any important amendments to the protocol will be recorded in PROSPERO record and discussed in the methods section of the final report.

### Eligibility criteria
We will include all randomised and quasirandomised controlled trials comparing the effect of intraoperative high $FiO_2$ (0.60–1.0) to traditional $FiO_2$ (0.21–0.40) in patients undergoing surgery. Definitions for high and low $FiO_2$ were determined by literature review and consensus among the IPDMA collaborators.[21 22] These trials may include patients of any age undergoing surgery except for neonates, regardless of publication, language or year of conduct and should include at least data on age, sex, mean $FiO_2$ administered, method of oxygen

administration, SSI, mortality or other serious adverse events. Any outcome found to be recorded in these trials will be included in the analysis. Studies without random or quasirandom treatment allocation, animal studies and studies outside of the intraoperative period will be excluded.

### Identifying studies: information sources

The initial search conducted for the WHO guideline will be updated by a professional information specialist.[21 22] Medical databases will be searched, including MEDLINE, Embase, CENTRAL, CINAHL, ClinicalTrials.gov and the WHO regional databases. Online trial registries will be searched to identify potential unpublished evidence or any ongoing trials. The search will not be limited by language or date of publication. A final update will be conducted before the final round of revisions preceding submission for publication. The reference list of all included studies will be hand searched for any additional relevant trials, not already identified through database searching. All corresponding authors of relevant clinical trials will be contacted to review the list of identified studies for the omission of potentially relevant studies missed by the search.

### Study selection process

Two reviewers will independently assess articles retrieved by the search against the eligibility criteria. After screening the title and abstract using Rayyan, the full text of potentially eligible papers will be retrieved and assessed.[36] When no full paper exists, or trial eligibility is in doubt, the study authors will be contacted to provide further information. Any discrepancies in study selection will be resolved through consensus and discussion with a senior author. All studies that pass title and abstract screening but were not eligible for inclusion will be listed with the reasons for exclusion.

### Study collaboration invitation

Authors of eligible studies will be contacted and invited to collaborate on the IPD MA. An email invitation will be sent to the corresponding authors outlining the IPD MA goals. If no reply is received within 2 weeks, a second email request will be sent to the corresponding and first author. If no response is received again, we will try to contact all authors by email and telephone. IPD data will be considered unavailable if numerous times (at least five) no reply is received and if authors no longer have access to the study data or consent to collaboration.

### Data collection process

The collaborating investigators will be requested to sign a data transfer agreement describing the ownership and storage of the IPD before IPD is shared. Whenever possible, data collection, interview on the protocol and formal handoff on the data codebook will be done electronically via email, videoconference or a suitable alternative. Whenever requested by the original investigator, a researcher will visit the investigators for a physical data transfer, in-person interview and data codebook handoff. IPD will be deidentified by the suppling collaborator. The IPD deidentification code will not be shared. IPD will be transferred using one of the following secure methods: SurfFilesender, a secure password protected data transfer service,[37] end-to-end encrypted and password protected using email or sent by courier on a physical storage media. Once transferred, IPD will be stored securely on the local server of the Amsterdam UMC where appropriate data and privacy policies will be maintained, as well as procedures and associated physical, technical and administrative safeguards to assure that the IPD are accessed only by authorised personnel. In the unlikely event that IPD will not be made available, the reason will be recorded. The aggregate data of the study will be used in a sensitivity analysis. Aggregate data collection will be performed as appropriate for a regular meta-analysis by two independent reviewers according to a predefined data extraction sheet and overseen by a senior author to settle potential discrepancies. The University of Amsterdam's Clinical Research Unit will facilitate secure data storage.

### Data items

Data items will include all data recorded by the initial trial investigators including, but not limited to the items listed in tables 1 and 2. SSI within 90 days after surgery according to the authors' discretion will be the primary outcome, all-cause mortality within the longest available follow-up will be the secondary outcome. All other outcomes are exploratory.

### Missing data

When variables are missing at the participant level and the missing at random assumption is plausible, multiple imputations by chained equations may be applied in each trial separately before proceeding with the analysis. Variables that miss systematically, that is, unknown for the entire study or are deemed missing non-randomly after discussion in the writing committee, will not be imputed. When this concerns variables included in predefined analysis, studies systematically missing this variable will be excluded from that analysis. When this concerns variables not included in predefined analysis, these variables will be dropped from the main outcome analysis as potential confounding variables. The set of available variables for the main analysis will thus be determined by the data set with the least available variables. Variables from richer sets will remain available for exploratory analysis among data sets with the variable available.

### Individual participant data integrity

We will check IPD for potential missing, invalid, or out-of-range values, inconsistencies, and discrepancies with the aggregate publication. When identified, we will seek to resolve the issues with the trial investigators to improve data quality and ensure that trials are represented accurately. In addition, any modelling assumptions made in the initial analysis will be evaluated (ie, missing at random

**Table 1** Baseline and procedure characteristics

| | |
|---|---|
| Baseline | Sex, age, body mass index (kg/m$^2$), American Society of Anaesthesiologists (ASA) physical status score, smoking status, peripheral vascular disease, diabetes, (metastatic) cancer, congestive heart disease, (pulmonary) hypertension, chronic obstructive pulmonary disease, immunosuppressant use, peripheral oxygen saturation (%), glucose (mg/dL), indication for surgery, emergency procedure |
| Preoperative | Use of preoperative antibiotic prophylaxis (dose and agent), timing of preoperative antibiotic prophylaxis, use of mechanical bowel preparation, haemoglobin (g/L), use of antibiotic bowel preparation, cytostatic chemotherapy, radiotherapy, use of preoperative skin preparation prophylaxis (dose and agent), timing of preoperative skin preparation prophylaxis |
| Intraoperative | Surgical procedure(s), organ involvement, contamination (Centers for Disease Control and Prevention wound classification[61]), laparoscopic surgery, mean arterial pressure (mm Hg), haemoglobin (g/L), heart rate (beats/min), haemodynamic monitoring method, haemodynamic management algorithm, crystalloid infusion (mL), colloid Infusion (mL), red cell transfusion (units), duration of surgery (min), duration of anaesthesia (min), mean core temperature (°C)*, lowest core temperature (°C)*, duration hypothermia (<35°C), mean net fluid supplementation (mL/kg/hour), arterial oxygen saturation (%), peripheral oxygen saturation (%), subcutaneous oxygen tension (mm Hg), muscle oxygen tension (mm Hg), partial pressure of arterial oxygen (mm Hg), mean $FiO_2$ (%), mean positive end-expiratory pressure (PEEP) ($cmH_2O$), ventilator flow (L/min), peak airway pressure (mm Hg), plateau pressure (mm Hg), tidal volume (mL/kg predicted body weight), respiratory frequency, vasopressor agent, vasopressor agent used (mg), glucose (mg/dL), use of general anaesthesia, use of spinal or epidural anaesthesia, use of mechanical ventilation, use of nitrous oxide, total blood loss (mL), fluids (mL), end tidal $CO_2$ |
| Postoperative | Use of postoperative antibiotics (dose and agent), postoperative antibiotic duration (days), cytostatic chemotherapy, radiotherapy, haemoglobin (g/L), visual analog scale (VAS) pain score, use of postoperative oxygen suppletion (duration, method and $FiO_2$), haemoglobin (g/L), peripheral oxygen saturation (%), partial pressure of arterial oxygen (mm Hg), subcutaneous oxygen tension (mm Hg), muscle oxygen tension (mm Hg), glucose (mg/dL), National Nosocomial Infections Surveillance System (NNIS) Score,[62] surgical site infection risk score,[63] use of drains |
| Oxygen administration and monitoring | Total duration and concentration of oxygen exposure during the preoperative/intraoperative/postoperative period (timing of initiation, concentration, duration), oxygen supply and mode of administration (intubation, use and type of face mask, nasal prongs), carrier gas ($N_2$, $N_2O$, medical or room air), protocol-defined target or range of partial pressure of arterial oxygen (mm Hg) or peripheral oxygen saturation (%) |

*Direct measurement or its approximation by peripheral measurement.

**Table 2** Outcome data and effect measure specification

| | |
|---|---|
| **Primary** | **SSI within 90 days after surgery by the author's discretion*** |
| Secondary | All-cause mortality within the longest available follow-up |
| Exploratory | ▶ Survival within the longest available follow-up <br> ▶ Serious adverse events defined by the International Council on Harmonisation of Technical Requirements for Registration of Pharmaceuticals for Human Use (ICH) guidelines for good clinical practice[64] <br> ▶ SSI monitored according to the Centers for Disease Control and Prevention criteria and specified as either superficial, deep, organ/space[65] <br> ▶ Respiratory insufficiency: defined as the need for respiratory assistance provided as ventilator therapy or non-invasive ventilation within 90 days after surgery <br> ▶ Unplanned intensive care unit admission (not part of routine postoperative care) (days) <br> ▶ Hospital readmissions within 90 days after surgery <br> ▶ Anastomotic leakage as defined by the international study group of rectal cancer[66] <br> ▶ Total duration of hospitalisation, including readmissions related to the initial hospitalisation <br> ▶ Any cardiovascular complication at any time after surgery <br> ▶ Any pulmonary complications at any time after surgery <br> ▶ Stroke at any time after surgery <br> ▶ New or recurrent cancer diagnosis at any time after surgery <br> ▶ Any further clinically relevant outcome reported in the individual participant data |

*When patients are reoperated within follow-up for reasons other than surgical site infection, these cases will be excluded from the analysis based on loss to follow-up.
SSI, surgical site infection.

de Jonge SW, *et al. BMJ Open* 2023;**13**:e067243. doi:10.1136/bmjopen-2022-067243

in the case of multiple imputation or non-informative censoring and proportional hazards in the case of time to event data). In the case of any concerns on IPD integrity, further prove of execution of the trial and substantiation of the results may be requested such as prove of institutional review board approval or original case record forms. If concerns cannot be resolved with the trial investigators, the data of the concerning study will not be included in the primary analysis and the reason for exclusion will be explicitly stated.

### Risk of bias

Two reviewers will independently assess the quality of the included studies using the Cochrane risk-of-bias tool for randomised trials V.2.[38] Studies will be judged as '*low risk*', '*some concerns*' or '*high risk* of bias'. Publication bias will be assessed using a contour enhanced funnel plot.[39] Additionally, the IPD will be used to directly check process parameters of some of the bias domains. Randomisation and allocation concealment will be assessed by checking baseline imbalances. Incomplete outcome data will be assessed by checking the IPD to ensure all randomised patients are included. All available clinically relevant outcomes in the IPD will be reported in the IPD MA. For time to event outcomes such as mortality, the pattern and extent of follow-up will be checked. When needed, additional follow-up with original authors will be conducted to rectify any imbalances as far as possible.

### Synthesis methods

All outcomes will be analysed according to the intention-to-treat principle and using a one-step approach for IPD MA. In the one-step approach, IPD will be modelled from all studies simultaneously using the generalised linear mixed model framework and the statistical model appropriate for the type of outcome being analysed (ie, logistic regression for binary outcome data, linear regression for continuous outcome data and Cox regression for time to event data). A random treatment effect term will be added to the model and all other parameters (intercepts, prognostic factor effects and residual variances) will be stratified by trial to account for the clustering of patients within studies. Maximum likelihood with quadrature will be used as estimation method and study-specific centreing of the variables.[40] Variables potentially affecting the outcome that, despite randomisation, show baseline imbalances across treatment arms will be considered for adjustment based on the criteria for confounder selection by VanderWeele and Shpitser.[41–44] Procedure duration is considered an important proxy for the complexity of the procedure and will also be considered for adjustment despite being measured during the exposure.[43 44] We assume that the $FiO_2$ used does not affect procedure duration. Benjamini-Hochberg correction will be used to account for multiple testing for the primary and secondary outcomes when appropriate.[45]

### Exploration of variation in effects

To explore the causes of heterogeneity and identify factors modifying the effects of high intraoperative $FiO_2$, we will perform prespecified subgroup analyses by extending the one-step meta-analysis framework to include treatment-covariate interaction terms. Subgroups will be defined according to mean core temperature (<35°C), mean net fluid supplementation (<15 mL/kg/hour), use of mechanical ventilation, use of nitrous oxide, use of preoperative antibiotic prophylaxis and procedure duration (>2.5 hour). All subgroup variables have been proposed as effect modifiers in previous studies and have a plausible biological substantiation.[46–57] Cut-offs are driven by previously reported data.[46–57] Treatment-covariate interaction terms p<0.05 will be considered statistically significant. Dose–response variation will be explored by total $O_2$ exposure duration for each primary outcome. All exploratory analysis will be interpreted with caution considering the limited power and potential of type 1 error when multiple interactions are tested.

### Additional analysis

A sensitivity analysis will be conducted to test the impact of excluding trials using $N_2O$ as a carrier gas on the pooled effect estimate. Further, the choice of SSI definition will be evaluated in a sensitivity analysis applying the CDC definition as the primary outcome. In the case of exclusion of trials due to concerns on IPD integrity, a sensitivity analysis will be conducted to test the impact of including the concerning data on the pooled effect estimate. When multiple imputation is performed, a complete case analysis will also be conducted. In studies with sufficiently detailed data on the intervention, all analyses will also be conducted according to the per-protocol principle after adjustment for confounding factors due to incomplete adherence to the assigned treatments or use of off-protocol concomitant therapies according to the variable selection principles described for the primary analysis. Per-protocol treatment will be defined as an $FiO_2$ of 0.80±0.05 for at least 75% of the ventilation time in the intervention group, and an $FiO_2$ smaller than 0.40 with a margin of 0.05, for 75% in the control group. Patients requiring more oxygen for medical reasons, for example, to maintain adequate saturation, after initial ventilation with an $FiO_2$ of 0.45 are exempted and not considered a protocol deviation. A sensitivity analysis will be conducted according to the two-step approach. All studies will be reanalysed separately, similarly to the one-step approach but without the term for trial clustering. The new aggregate data of each study will then be synthesised in a second step synthesising an overall estimate using the maximum likelihood method followed by the Hartung-Knapp-Sidik-Jonkman correction assuming random effects.[58] Between-study variance will be evaluated using $\tau^2$; in addition, the $\chi^2$ test for heterogeneity will be performed with p<0.100 considered statistically significant. In the unlikely event that IPD will not be made available, aggregate study data will be included in the analyses during

step 2. Any unforeseen challenge during the analysis or choice that leads to discussion in the steering group that cannot be resolved by consensus will also be subjected to sensitivity analysis. To assess robustness of the time to event outcomes, a survival curve will be compared with the univariable version of the Cox proportional hazards regression analysis.

### Certainty of the cumulative estimate

The Grading of Recommendations, Assessment, Development and Evaluation working group methodology will be used to assess the overall quality of evidence for the following domains: risk of bias, unexplained inconsistency, indirectness, imprecision, publication bias and magnitude of effect. Additional domains may be considered where appropriate. Optimal information size, defined as the number of participants needed for a single adequately powered trial, was calculated assuming a type 1 error (α) of 0.05, a type 2 error (β) of 0.2 and a relative risk reduction of 0.25.[59] If a CI fails to exclude appreciable benefit or harm, defined as a relative risk reduction or increase of 0.25, the quality of evidence will be downgraded regardless of the optimal information size.[59] The overall certainty will be classified using four levels: high, moderate, low and very low.[60]

### Software

Results will be processed using R V.4.0.4.

## ETHICS AND DISSEMINATION
### Ethical approval

Because this concerns a study on existing deidentified patient data, the medical research involving human subjects act does not apply and no formal medical ethics review is required.

### Dissemination

This protocol and the results of this study will be submitted to a peer-reviewed medical journal regardless of the outcome. The protocol will be submitted before the data is gathered and analysed.

### Author affiliations
[1]Department of Surgery, Amsterdam UMC location University of Amsterdam, Amsterdam, The Netherlands
[2]Department of Anaesthesiology, Amsterdam UMC location University of Amsterdam, Amsterdam, The Netherlands
[3]Amsterdam Gastroenterology Endocrinology and Metabolism, Amsterdam, The Netherlands
[4]Department of Outcomes Research, Cleveland Clinic, Cleveland, Ohio, USA
[5]Department of Anaesthesia and Intensive Care, Copenhagen University Hospital - Bispebjerg and Frederiksberg, Copenhagen, Denmark
[6]Department of Critical Care, Melbourne Medical School, University of Melbourne, Melbourne, Victoria, Australia
[7]Department of Anaesthesiology and Perioperative Medicine, Alfred Hospital, Monash University, Melbourne, Victoria, Australia
[8]Department of Epidemiology and Preventive Medicine, Monash University, Melbourne, Victoria, Australia
[9]Department of Anaesthesiology and Pain Medicine, Bern University Hospital, University of Bern, Bern, Switzerland
[10]Department of Anaesthesiology and Critical Care Medicine, Johns Hopkins University, Baltimore, Maryland, USA
[11]Department of General Anaesthesiology, Cleveland Clinic, Cleveland, Ohio, USA
[12]Department of Anaesthesiology and Perioperative Medicine, and Department of Epidemiology and Biostatistics, University of Western Ontario, London, Ontario, Canada
[13]Epidemiology and Data Science, Amsterdam UMC, University of Amsterdam, Amsterdam, The Netherlands
[14]Amsterdam Public Health, Methodology, Amsterdam UMC location University of Amsterdam, Amsterdam, The Netherlands
[15]Department of Anaesthesiology, Weil Medical College of Cornell University, New York City, New York, USA
[16]Department of Surgery, Hospital Clinico Universitario de Valencia, Valencia, Valenciana, Spain
[17]Department of Anaesthesia and Critical Care, Hospital Clinico Universitario de Valencia, Valencia, Spain
[18]Department of Anaesthesiology and Critical Care, Hospital Clínic de Barcelona, Barcelona, Spain
[19]CIBER de Enfermedades Respiratorias, Instituto de Salud Carlos III, Madrid, Spain
[20]Department of Anaesthesiology and Critical Care Medicine, Ben-Gurion University of the Negev, Be'er Sheva, Israel
[21]Department of Obstetrics and Gynaecology, Indiana University School of Medicine, Indianapolis, Indiana, USA
[22]Department of Obstetrics and Gynaecology, Wright State University and Miami Valley Hospital, Dayton, Ohio, USA
[23]Department of Anaesthesia and Intensive Care, The Chinese University of Hong Kong, Hong Kong, China
[24]Centre d'appui pour la Prévention des Infections Associées aux Soins, CPias Normandie, Centre Hospitalo-Universitaire, Caen, Normandy, France
[25]World Federation of Societies of Anesthesiologists, London, UK
[26]World Health Organization, Geneva, Switzerland

**Acknowledgements** The authors thank Faridi Jamaludin as professional information specialist for the search.

**Contributors** SWdJ conceived the study. SWdJ, MWH, MB and RHH secured funding for the study. SWdJ, MWH, MB, MGWD and RHH designed the study, drafted the study protocol and provided statistical expertise. RHH and MZN provided input for the literature search and will coordinate the assembly of the data and perform the screening, inclusion and assessment of risk of bias. RPW, CSM, KL, PM, AF, RG, OA, AK, DS, JM, KP, FJB, CF, GMG, CMS, DSM, MTVC, PT, JM-O and BA provided substantial contributions to the study design, provided critical feedback and approved the final version of the study protocol.

**Funding** This work was supported by the Amsterdams Universiteitsfonds grant (4069) to RHH.

**Disclaimer** The author is a staff member of the World Health Organization. The author alone is responsible for the views expressed in this publication and they do not necessarily represent the views, decisions or policies of the World Health Organization. The author is a staff member of the World Health Organization. The author alone is responsible for the views expressed in this publication and they do not necessarily represent the views, decisions or policies of the World Health Organization.

**Competing interests** SWdJ reports receipt of grants from Photonics in Healthcare, Integra LifeSciences and Ethicon outside the submitted. PM reports receipt of grants or contracts from the Australian National Health and Medical Research Council (NHMRC), Practitioner Fellowship and Projects Grants, payment of expert testimony from Avant Medical Indemnity, and participation on a Data Safety Monitoring Board of Advisory Board for the SNAP, TOPIC-2 and BONANZA trials. AF reports receipt of institutional grants from the Australian Research Council Discovery Project and National Health and Medical Research Council Ideas outside the submitted work and participation on a Data Safety Monitoring Board or Advisory Board for the Australian Kidney Trials Research Network (INCH-HD, IMPEDE, TEQCH-PD, PHOSPHATE, BEST Fluids, N3RO trial, CKD-FIX, IMPROVE-FIX). RG reports participation on Steering Committee for the IntuBot Innosuisse Projekt and has a leadership role as ERC Director of Guidelines and ILCOR, and ILCOR Task Force Chair on Education, Implementation and Team, and Treasurer of European Airway Management society, and reports receipt without any payment of airway

equipment for the research of the following: Intersurgical, Karl Storz, Verathon, Aircraft Medical, Prodol Meditec, Venner Medical, Kingsystems, Medtronic, Ambu, VBM, Radiometer, Sentec and Fisher & Paykel. AK reports receipt of grants or contracts from Potrero Medical, Rehabtronics and The 37Company outside the submitted work and participation on a Data Safety Monitoring Board of Advisory Board in Directed systems, Potrero Medical and BioAgel Laboratories. JM-O reports voluntary participation as a panellist in the updated WHO Guidelines on high versus low FiO2 in 2018 and voluntary coinvestigator in the PENGUIN trial of high versus low FiO2 for SSI prevention in abdominal surgery in low-income and middle-income settings with GlobalSurg Collaborative. MGWD reports participation on a Data Safety Monitoring Board or Advisory Board for the following trials: DANCE, SPHINX, ICONIC, SAFE, PACER, LEARNS, RECAP and BIOPEX2. CF reports receipt fees for lectures and educational events from Getinge and Medtronic outside the submitted work. CMS reports receipts of institutional grants from NICHD outside the submitted work. MB reports receipt of institutional grants from KCl/3M, Johnson & Johnson, New Compliance, BD Bard, Gore, Telabio, GDM, Medtronic and Smith & Nephew outside the submitted work, and participation on the Data Monitoring Committee of the EXTEND trial. MWH reports receipt of institutional grants from ZonMw outside the submitted work, consulting institutional fees from IDD Pharma outside the submitted work, institutional payment or honoraria for lectures, presentations, speakers bureaus, manuscript writing or educational events from CSL Behring outside the submitted work and has a leadership role in DGAI, ISAP and IARS (Anaesthesia and Analgesia). The other authors declare no conflict of interest.

**Patient and public involvement** Patients and/or the public were not involved in the design, or conduct, or reporting, or dissemination plans of this research.

**Patient consent for publication** Not applicable.

**Provenance and peer review** Not commissioned; externally peer reviewed.

**ORCID iDs**
Stijn W de Jonge http://orcid.org/0000-0003-0693-3111
Rick H Hulskes http://orcid.org/0000-0002-2153-3514
Paul Myles http://orcid.org/0000-0002-3324-5456
Ozan Akca http://orcid.org/0000-0002-7275-1060
Marcel GW Dijkgraaf http://orcid.org/0000-0003-0750-8790
Kane Pryor http://orcid.org/0000-0002-9212-5526

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
