## [Reviewer comments · BMJ Open]

ARTICLE DETAILS

TITLE (PROVISIONAL)	The benefits and harms of perioperative high fraction inspired oxygen for surgical site infection prevention: a protocol for a systematic review and meta-analysis of individual patient data of randomised controlled trials
AUTHORS	de Jonge, Stijn; Hulskes, Rick; Zokaei Nikoo, Maedeh; Weenink, Robert; Meyhoff, Christian; Leslie, Kate; Myles, Paul; Forbes, Andrew; Greif, Robert; Akca, Ozan; Kurz, Andrea; Sessler, Daniel; Martin, Janet; Dijkgraaf, Marcel; Pryor, Kane; Belda, F. Javier; Ferrando, Carlos; Gurman, Gabriel; Scifres, Christina; McKenna, David; Chan, MTV; Thibon, Pascal; Mellin-Olsen, Jannicke; Allegranzi, B; Boermeester, Marja; Hollmann, Markus

VERSION 1 – REVIEW

REVIEWER	Cumpstey, Andrew F. University Hospital Southampton NHS Foundation Trust
REVIEW RETURNED	05-Oct-2022

GENERAL COMMENTS	Thank you very much for the opportunity to review this work. This is a very well written and thought out project that will address an urgent and important question in perioperative research in an extremely thorough and detailed way. I wish the authors well in completing what is no doubt an ambitious project and large amount of work, and very much look forward to seeing their results soon.
---

REVIEWER	Oliveira, Ramon Universidade de São Paulo, Escola de Enfermagem
REVIEW RETURNED	18-Dec-2022

GENERAL COMMENTS	Thanks for the opportunity to review this interesting SR protocol. I have a few questions/suggestions, as follows: - The type and timing of skin prep should be included in the preoperative data and the use of drains should be included in the postoperative data (Table 1). These are important variables in the pathogenesis of SSIs.- It is very intriguing why 'SSI within 90 days after surgery by the authors' discretion' was selected as the primary outcome instead of the CDC's definition which is internationally accepted. The meta-analysis results would be more reliable if an accepted definition would be employed.- The core body temperature will, probably, not be available for a broad range of patients. Then, will the authors consider using the temperature recorded at the surface level?- How 'high FiO2 and Low FiO2' were determined?
---

	- Authors should explain how the data protection strategies will be implemented as well as who will be in charge of patient data de-identification.
--	---

REVIEWER	Pandit, Jaideep University of Oxford, Nuffield Department of Anaesthetics
REVIEW RETURNED	28-Dec-2022

GENERAL COMMENTS	This is a very relevant study proposal from a wide, representative and well qualified author group. The methods are ambitious but with the extensive collaborations already made, there should a high chance of success. It will be very interesting indeed to see how or in what ways this approach produces different results from traditional meta-analyses. My queries are only minor: 1. ethics: the authors state that ethical approval is not needed, but I felt some more information was needed on this point, eg, what advice they had sought etc. The study is an international ones, with different ethical processes for the original studies, and a question is to what extent such further patient-level analysis is permitted vs pooled analysis. I do not see an objection, but perhaps a short sentence or reference? 2. I note the 2 levels of O2 which clearly leaves a gap (0.41-0.59). I suspect this is determined by original studies if none studied this gap. But a thought is what this implies for clinical practice? If high level found beneficial is it then inappropriate to use say 0.5 (which I suspect is a common or even modal choice)? Perhaps not germase for this methods paper. Also I was surprised to see 0.21 which is quite low, but again I suspect this is determined by original studies. Methods papers are always difficult to comment on, as things like statistics cannot really be commented on without reference to some data. Also, it is conceivable that aspects of methods may change based on the exact data that reveals itself. Nevertheless this is a really interesting approach to this problem.
---

REVIEWER	Apte, Sameer Ottawa Hospital, Surgical Oncology
REVIEW RETURNED	08-Jan-2023

GENERAL COMMENTS	This will be a very important paper published, largely because the IPD MA seems to be well thought out and has not been previously performed. I think the interpretation of the results will be critical, as the paper does attempt to analyse a large number of outcomes and might suffer slightly from the problem of multiple comparisons - This is to be expected with an exploratory analysis. Overall, I think this should be published. See below for my specific comments. Thank you for doing this important work. Notes: Overall – a paper with a strong statistical methodology, but the writing could use some more detail and explanation of the rationale. Abstract Intro–
---

	The Intro could use some specific examples. For instance, the authors say 'perioperative care has changed considerably with consequences for'... 'These changes may explain'. These sentences assume that the reader has a background in perioperative care, and would know inherently what changes the authors are talking about. Specific example or two of the changes in question would help. Similarly, 'inconsistency in results' – The reader is not made aware of what results the authors refer to. Were there two similar papers – one showing no decrease in SSI with high FiO2 use, and another null effect? Or was it two meta-analyses/sys reviews with disparate conclusions? Using more examples would strengthen the abstract introduction Abstract Methods – Would specific some examples of which medical databases and trial registries. Effect of 'high' and 'regular' FiO2 on which outcome? Is it the effect on SSI? Or the effect on another outcome? Will the analysis use difference in proportions? Odds ratios? Hazard ratios? The outcome should be explicit in the methods. Additionally, the specific methods used should be outlined. What type of meta-analysis is this going to be? It would not be appropriate to simply pool all the data and analyse it as a single randomised controlled trial – These patients will come from different populations with heterogeneity in trial design and patients. This needs to be taken into account when performing the meta-analysis, and there are no specific details on the statistical methods used. (i.e. fixed vs random-effects, or main meta analytical outcome measure used, method for pooling heterogeneous datasets etc.) Analyzing effect modifiers is commendable, but due to the severe reduction in study power, one (or very rarely) two a priori effect-modification hypotheses should be stated, and adhered to. It is not clear what effect modification the authors are looking for. I worry that if too many variables are analysed for effect modification this will lead to two issues 1) the problem of multiple comparisons, and 2) severe reduction in study power (and inflation of confidence intervals) due to including too many coefficients in your regression model. Article summary (strengths and limitation) While IPD MA is a very strong method, I think it is probably a bit controversial to say it is the gold standard. An well-powered individually randomised trial across multiple continents and health care centre-types investigating a specific surgical patient population would give a more reliable result for that specific patient population than an IPD MA of heterogeneous trial in my opinion. Intro – Similar to abstract, there are a few places where examples would be helpful – 'Despite many randomised controlled trials, various meta-analyses, and guideline recommendations, uncertainty remains.' Uncertainty in what outcome or clinical concern specifically? What patient populations? I see that there is more examples and info in the body introduction in how perioperative care has changed which is great to see. Again in regards to the primary outcome – it is not explicitly stated that the authors are investigating the effect of high FiO2 on SSI rate reduction. Any further possible analyses of variables that might be affected by high FiO2 should be considered strictly exploratory, or secondary. Methods: I see now that the authors do have clearly outlined primary, secondary and exploratory analyses in the methods. I think all that
--	---

	needs to be done is tweak the introduction and methods sections to relay this information better in the text. I also think that the section on analysis of missing data, and using multiple imputations for non-systematically missing data is a strength. It would be nice to see how the authors will determine if data is missing systematically – will they do a significance test of the missing data against the outcome in question? (not a critical critique, but should be outlined clearly in the final analysis). Synthesis methods – It is also now obvious that the authors have defined a relatively clear methodology for systematic review analysis. I think some of this information could be included in the abstract to give the readers a better idea what is to be done, without reading the whole protocol. I also see that the authors have considered heterogeneous patient publications and are including a random-effects terms – again would nice if this was in the abstract to make this clear to the reader that the analysis methods are sound (they do seem to be). Exploration of variation in effects – It is good to see the specific variables considered for interaction (effect modification) are listed here. I still do have a bit of a concern that this will cause a problem of multiple comparisons. The authors should also address (or at least mention) the difficulties in performing too many analyses for effect modification. I understand that at this stage, it will highly depend on the type/quality of data received, but would be nice to mention. Also, it might be nice if some of this information could be included in the abstract to help a cursory reader see that the statistical plan is in fact well-thought out. Additional analysis. Very nice to see complete-case analysis being used as well as MI – consistency of results will lend more strength to the conclusions. The methods of using one versus two-step IPD seem sound, and both account for study heterogeneity and random effects. For the very last line of this section – Will the survival curve be a univariable meta-analysed KM curve, or will this be a quasi-controlled survival curve derived from Cox regression point estimates? Either one would be fine I think. Competing Interests – Use of GRADE is a strength
--	--

VERSION 1 – AUTHOR RESPONSE

Reviewer 1	
Comment #1	"Thank you very much for the opportunity to review this work. This is a very well written and thought out project that will address an urgent and important question in perioperative research in an extremely thorough and detailed way. I wish the authors well in completing what is no doubt an ambitious project and large amount of work, and very much look forward to seeing their results soon."
Response	We thank the reviewer for the careful review of our manuscript and for these kind words.
Reviewer 2	
General response	We thank the reviewer for the careful review of our manuscript and for these thoughtful suggestions. Please find any potentially relevant references at the bottom of the document.

Comment #1	The type and timing of skin prep should be included in the preoperative data and the use of drains should be included in the postoperative data (Table 1). These are important variables in the pathogenesis of SSIs.
Response	We have added the suggested variables. Table 1. Baseline and procedure characteristics, pages 10: "Use of preoperative skin preparation prophylaxis (dose and agent), Timing of preoperative skin preparation prophylaxis "and "Postoperative drains".
Comment #2	It is very intriguing why 'SSI within 90 days after surgery by the authors' discretion' was selected as the primary outcome instead of the CDC's definition which is internationally accepted. The meta-analysis results would be more reliable if an accepted definition would be employed.
Response	The CDC's definition is included in the protocol as a secondary outcome to accommodate comparability to literature on other interventions. The longer follow up and definition according to the author's discretion were chosen as primary endpoint to optimize utilisation of the available data. Not all studies have used the CDC definition. It was not as widespread known as it is now, at the time of the early research on this topic. To address any concern that this choice may affect the outcome, we will conduct a sensitivity analysis. Methods and analysis, Additional analysis, page 13, lines 289-290: "Further, the choice of SSI definition will be evaluated in a sensitivity analysis applying the CDC definition as the primary outcome."
Comment #3	The core body temperature will, probably, not be available for a broad range of patients. Then, will the authors consider using the temperature recorded at the surface level?
Response	Indeed, core temperature or its approximation by peripheral measurement will be considered. This nuance is specified. Table 1. Baseline and procedure characteristics, intraoperative, page 10: "....mean core temperature (□C)*, lowest core temperature (□C)*..." * Direct measurement or its approximation by peripheral measurement
Comment #4	How 'high FiO2 and Low FiO2" were determined?
Response	High and low FiO2 were determined by literature review ^{1,2} and consensus among the collaborators. We have added this explanation to the text. Methods and analysis. Eligibility criteria. Page 8, Line 160-161: Definitions for high and low FiO₂ were determined by literature review and consensus among the IPDMA collaborators.^{1,2}
Comment #5	Authors should explain how the data protection strategies will be implemented as well as who will be in charge of patient data de-identification."
Response	Specification of data protection strategies and responsibilities regarding de-identification have been added to the text Data collection process, page 9, lines 201-207:

	"IPD will be de-identified by the suppling collaborator. The IPD de-identification code will not be shared. IPD will be transferred using one of the following secure methods: SurfFilesender, a secure password protected data transfer service,³ end-to-end encrypted and password protected using email or send by courier on a physical storage media. Once transferred, IPD will be stored securely on the local server of the Amsterdam UMC where appropriate data and privacy policies will be maintained, as well as procedures and associated physical, technical and administrative safeguards to assure that the IPD are accessed only by authorized personnel."
Reviewer 3	
General response	We thank the reviewer for the careful review of our manuscript, kind words and for these thoughtful suggestions. Please find any potentially relevant references at the bottom of the document.
Comment #1	Ethics: the authors state that ethical approval is not needed, but I felt some more information was needed on this point, e.g., what advice they had sought etc. The study is an international ones, with different ethical processes for the original studies, and a question is to what extent such further patient-level analysis is permitted vs pooled analysis. I do not see an objection, but perhaps a short sentence or reference?
Response	Wording is added to specify why no formal medical ethics review is required. Ethics and dissemination, Ethical approval, page 14, lines 331-332: "Because this concerns a study on existing de-identified patient data, the medical research involving human subjects act does not apply and no formal medical ethics review is required."
Comment #2	I note the 2 levels of O2 which clearly leaves a gap (0.41-0.59). I suspect this is determined by original studies if none studied this gap. But a thought is what this implies for clinical practice? If high level found beneficial is it then inappropriate to use say 0.5 (which I suspect is a common or even modal choice)? Perhaps not germase for this methods paper. Also I was surprised to see 0.21 which is quite low, but again I suspect this is determined by original studies.
Response	Indeed, high, and low FiO₂ were determined by literature review^{1, 2} and consensus among the collaborators. We have added this explanation to the text. The reviewer rightfully points out that this leaves a gap between the defined groups and that the groups include very low and very high FiO₂ values. Depending on the data collected and the results of the analyses, any remaining knowledge gaps will be addressed in the discussion section of the final study

	report. A dose response variation will be explored by total O₂ exposure duration for each primary outcome. Methods and analysis. Eligibility criteria, page 8, Line 160-161: “Definitions for high and low FiO₂ were determined by literature review and consensus among the IPDMA collaborators.^{1, 2”} Methods and analysis. Exploration of variation in effects, page 13, lines 283-284: “Dose-response variation will be explored by total O₂ exposure duration for each primary outcome.”
Reviewer 4	
General response	We thank the reviewer for the careful review of our manuscript, kind words and for these thoughtful suggestions. Please find any potentially relevant references at the bottom of the document.
Comment #1	The Intro could use some specific examples. For instance, the authors say 'perioperative care has changed considerably with consequences for'... 'These changes may explain'. These sentences assume that the reader has a background in perioperative care, and would know inherently what changes the authors are talking about. Specific example or two of the changes in question would help. Similarly, 'inconsistency in results' – The reader is not made aware of what results the authors refer to. Were there two similar papers – one showing no decrease in SSI with high FiO₂ use, and another null effect? Or was it two meta-analyses/sys reviews with disparate conclusions? Using more examples would strengthen the abstract introduction."
Response	Wording is added to strengthen the abstract introduction in line with the suggestions. When appropriate, corresponding modifications have been added to the main introduction. Abstract, introduction, page 3, lines 50-55: “Promising results of early randomized controlled trials (RCT) have been replicated with varying success and subsequent meta-analysis are equivocal. Recent advancements in perioperative care, including the increased use of laparoscopic surgery and pneumoperitoneum and shifts in fluid and temperature management, can affect peripheral oxygen delivery and may explain the inconsistency in reproducibility. However, the published data provides insufficient detail on the participant level to test these hypotheses.” Introduction, page 5, lines 95-98: “Concerns were raised on the safety of the use of high FiO₂ as well as on the conflicting study results with some in support of the

	use of high FiO₂ to reduce SSI and some not.⁴⁻¹⁴ Finally, studies by of one of the authors that contributed to the body of evidence were retracted because of unreproducible statistics.”
Comment #2	Would specific some examples of which medical databases and trial registries. Effect of 'high' and 'regular' FiO₂ on which outcome? Is it the effect on SSI? Or the effect on another outcome? Will the analysis use difference in proportions? Odds ratios? Hazard ratios? The outcome should be explicit in the methods. Additionally, the specific methods used should be outlined. What type of meta-analysis is this going to be? It would not be appropriate to simply pool all the data and analyse it as a single randomised controlled trial – These patients will come from different populations with heterogeneity in trial design and patients. This needs to be taken into account when performing the meta-analysis, and there are no specific details on the statistical methods used. (I.e. fixed vs random-effects, or main meta analytical outcome measure used, method for pooling heterogeneous datasets etc.) Analysing effect modifiers is commendable, but due to the severe reduction in study power, one (or very rarely) two a priori effect-modification hypotheses should be stated, and adhered to. It is not clear what effect modification the authors are looking for. I worry that if too many variables are analysed for effect modification this will lead to two issues 1) the problem of multiple comparisons, and 2) severe reduction in study power (and inflation of confidence intervals) due to including too many coefficients in your regression model."
Response	Wording is added to strengthen the method and analysis section of the abstract in line with the suggestions. Due to the limited word count of the abstract some details will only be addressed in the full methods section of the main text. Six pre-specified effect modifiers will be explored independently and interpreted with caution as appropriate in accordance with the concerns of the reviewer. This is detailed in the main text. Abstract, methods and analysis, page 3 lines 60-68: “Two reviewers will search medical databases and online trial registries, including MEDLINE, EMBASE, CENTRAL, CINHALL, and clinicaltrial.gov, for randomised and quasi randomised controlled trials comparing the effect of intraoperative high FiO₂ (0.60-1.00) to regular FiO₂ (0.21-0.40) on SSI within 90 days after surgery in adult patients. Secondary outcome will be all-cause mortality within the longest available follow-up. Investigators of the identified trials will be invited to collaborate, comment on the study protocol, and supply the individual participant data of their initial trial and additional follow-up data. Data will be analysed with the one step approach using the generalised linear mixed model framework and the statistical model appropriate for the type of outcome being analysed, with a random treatment effect term to account for the clustering of patients within studies. The certainty of evidence will be assessed using GRADE methodology.”

	Methods and analysis, exploration of variation in effects, Pages 12-13, lines 275-284: “To explore the causes of heterogeneity and identify factors modifying the effects of high intraoperative FiO₂, we will perform pre-specified subgroup analyses by extending the one-step meta-analysis framework to include treatment-covariate interaction terms. Subgroups will be defined according to mean core temperature (<35°C), mean net fluid supplementation (<15ml/kg/hr), use of mechanical ventilation, use of nitrous oxide, use of preoperative antibiotic prophylaxis, and procedure duration (>2.5h). All subgroup variables have been proposed as effect modifiers in previous studies and have a plausible biological substantiation.¹⁵⁻²⁶ Cut-offs are driven by previously reported data.¹⁵⁻²⁶ Treatment-covariate interaction terms p <0.05 will be considered statistically significant. Dose-response variation will be explored by total O₂ exposure duration for each primary outcome.”
Comment #3	Article summary (strengths and limitation) While IPD MA is a very strong method, I think it is probably a bit controversial to say it is the gold standard. An well-powered individually randomised trial across multiple continents and health care centre-types investigating a specific surgical patient population would give a more reliable result for that specific patient population than an IPD MA of heterogeneous trial in my opinion.
Response	The wording regarding the gold standard is removed in accordance with the suggestion Article summary, page 4, lines 78-80 “Individual participant data meta-analysis (IPD MA) of (quasi-)randomised controlled trials provides the best possible analysis of the available data on the participant level, permitting the investigation of potential effect modifiers.”
Comment #4	"Intro – Similar to abstract, there are a few places where examples would be helpful – 'Despite many randomised controlled trials, various meta-analyses, and guideline recommendations, uncertainty remains.' Uncertainty in what outcome or clinical concern specifically? What patient populations? I see that there is more examples and info in the body introduction in how perioperative care has changed which is great to see. Again in regards to the primary outcome – it is not explicitly stated that the authors are investigating the effect of high FiO₂ on SSI rate reduction. Any further possible analyses of variables that might be affected by high FiO₂ should be considered strictly exploratory, or secondary.

	I see now that the authors do have clearly outlined primary, secondary and exploratory analyses in the methods. I think all that needs to be done is tweak the introduction and methods sections to relay this information better in the text.
Response	Wording is added to strengthen the introduction and methods section in accordance with the suggestions. Introduction, page 5, lines 103-105: “Despite various studies and recommendations, there is still no consensus on the safety and effectiveness of using high FiO₂ during surgery with regard to SSI, all-cause mortality and other adverse events in adult patients.” Introduction, page 6, lines 119-121: “The purpose of this IPD MA is to assess the potential benefits and harms of intraoperative high (0.60-1.00) FiO₂ compared to traditional (0.21-0.40) FiO₂ and its effect modifiers in adult patients undergoing surgery with SSI being the primary outcome.” Methods and analysis, Data items, table 2, pages 10-11, line 219: “SSI within 90 days after surgery according to the authors’ discretion will be the primary outcome, all-cause mortality within the longest available follow up will be the secondary outcome. All other outcomes are exploratory.”
Comment #5	I also think that the section on analysis of missing data, and using multiple imputations for non-systematically missing data is a strength. It would be nice to see how the authors will determine if data is missing systematically – will they do a significance test of the missing data against the outcome in question? (not a critical critique, but should be outlined clearly in the final analysis). Synthesis methods – It is also now obvious that the authors have defined a relatively clear methodology for systematic review analysis. I think some of this information could be included in the abstract to give the readers a better idea what is to be done, without reading the whole protocol. I also see that the authors have considered heterogeneous patient publications and are including a random-effects terms – again would nice if this was in the abstract to make this clear to the reader that the analysis methods are sound (they do seem to be). Exploration of variation in effects – It is good to see the specific variables considered for interaction (effect modification) are listed here. I still do have a bit of a concern that this will cause a problem of multiple comparisons. The authors should also address (or at least mention) the difficulties in performing too many analyses for effect modification. I understand that at this stage, it will highly depend on the type/quality of data received, but would be nice to mention. Also, it might be nice if some of this information could be included in the abstract to help a cursory reader see that the statistical plan is in fact well-thought out.

Response	Wording is added in accordance with the suggestions. Please also see responses to previous comments. To our knowledge, there is no statistical test for missing at random or missing systematically. This must be established by reason. Missing data items will be discussed in the writing committee to assess plausibility of the missing at random assumption. Methods and analysis, Missing data, page 11, line 224-225: “Variables that miss systematically i.e., unknown for the entire study or are deemed missing non-randomly after discussion in the writing committee, will not be imputed.” Methods and analysis, exploration of variation of effects, page 13, lines 284-285: “All exploratory analysis will be interpreted with caution considering the limited power and potential of type 1 error when multiple interactions are tested.”
Comment #6	"Additional analysis. Very nice to see complete-case analysis being used as well as MI – consistency of results will lend more strength to the conclusions. The methods of using one versus two-step IPD seem sound, and both account for study heterogeneity and random effects. For the very last line of this section – Will the survival curve be a univariable meta-analysed KM curve, or will this be a quasi-controlled survival curve derived from Cox regression point estimates? Either one would be fine I think."
Response	Wording has been added to clarify use of the Kaplan Meier curve. Methods and analysis, page 14, lines 310-312: “To assess robustness of the time to event outcomes a survival curve will be compared to the univariate version of the Cox proportional hazards regression analysis.”

1. de Jonge S, Egger M, Latif A, Loke YK, Berenholtz S, Boermeester M, Allegranzi B, Solomkin J. Effectiveness of 80% vs 30-35% fraction of inspired oxygen in patients undergoing surgery: an updated systematic review and meta-analysis. *Br J Anaesth* 2019;**122**(3): 325-334.
2. Mattishent K, Thavarajah M, Sinha A, Peel A, Egger M, Solomkin J, de Jonge S, Latif A, Berenholtz S, Allegranzi B, Loke YK. Safety of 80% vs 30-35% fraction of inspired oxygen in patients undergoing surgery: a systematic review and meta-analysis. *Br J Anaesth* 2019;**122**(3): 311-324.
3. SurfFilesender. <https://filesender.surf.nl>.
4. Hedenstierna G, Perchiazzi G, Meyhoff CS, Larsson A. Who Can Make Sense of the WHO Guidelines to Prevent Surgical Site Infection? *Anesthesiology* 2017;**126**(5): 771-773.

5. Volk T, Peters J, Sessler DI. The WHO recommendation for 80% perioperative oxygen is poorly justified. *Anaesthesist* 2017;**66**(4): 227-229.
6. Wenk M, Van Aken H, Zarbock A. The New World Health Organization Recommendations on Perioperative Administration of Oxygen to Prevent Surgical Site Infections: A Dangerous Reductionist Approach? *Anesth Analg* 2017;**125**(2): 682-687.
7. Meyhoff CS, Fonnes S, Wetterslev J, Jorgensen LN, Rasmussen LS. WHO Guidelines to prevent surgical site infections. *Lancet Infect Dis* 2017;**17**(3): 261-262.
8. Mellin-Olsen J, McDougall RJ, Cheng D. WHO Guidelines to prevent surgical site infections. *Lancet Infect Dis* 2017;**17**(3): 260-261.
9. Berrios-Torres SI, Umscheid CA, Bratzler DW, Leas B, Stone EC, Kelz RR, Reinke CE, Morgan S, Solomkin JS, Mazuski JE, Dellinger EP, Itani KMF, Berbari EF, Segreti J, Parvizi J, Blanchard J, Allen G, Kluytmans J, Donlan R, Schechter WP, Healthcare Infection Control Practices Advisory C. Centers for Disease Control and Prevention Guideline for the Prevention of Surgical Site Infection, 2017. *JAMA Surg* 2017;**152**(8): 784-791.
10. Solomkin J, Egger M, de Jonge S, Latif A, Loke YK, Berenholtz S, Allegranzi B. World Health Organization Responds to Concerns about Surgical Site Infection Prevention Recommendations. *Anesthesiology* 2018;**128**(1): 221-222.
11. Akca O, Ball L, Belda FJ, Biro P, Cortegiani A, Eden A, Ferrando C, Gattinoni L, Goldik Z, Gregoretti C, Hachenberg T, Hedenstierna G, Hopf HW, Hunt TK, Pelosi P, Qadan M, Sessler DI, Soro M, Senturk M. WHO Needs High FIO₂? *Turk J Anaesthesiol Reanim* 2017;**45**(4): 181-192.
12. Weenink RP, de Jonge SW, Preckel B, Hollmann MW. PRO: Routine hyperoxygenation in adult surgical patients whose tracheas are intubated. *Anaesthesia* 2020;**75**(10): 1293-1296.
13. Sperna Weiland NH, Berger MM, Helmerhorst HJF. CON: Routine hyperoxygenation in adult surgical patients whose tracheas are intubated. *Anaesthesia* 2020;**75**(10): 1297-1300.
14. Weenink RP, de Jonge SW, van Hulst RA, Wingelaar TT, van Ooij PAM, Immink RV, Preckel B, Hollmann MW. Perioperative Hyperoxyphobia: Justified or Not? Benefits and Harms of Hyperoxia during Surgery. *J Clin Med* 2020;**9**(3).
15. Pryor KO, Fahey TJ, 3rd, Lien CA, Goldstein PA. Surgical site infection and the routine use of perioperative hyperoxia in a general surgical population: a randomized controlled trial. *JAMA* 2004;**291**(1): 79-87.
16. Belda FJ, Aguilera L, Garcia de la Asuncion J, Alberti J, Vicente R, Ferrandiz L, Rodriguez R, Company R, Sessler DI, Aguilar G, Botello SG, Orti R, Spanish Reduccion de la Tasa de Infeccion Quirurgica G. Supplemental perioperative oxygen and the risk of surgical wound infection: a randomized controlled trial. *JAMA* 2005;**294**(16): 2035-2042.
17. Meyhoff CS, Wetterslev J, Jorgensen LN, Henneberg SW, Hogdall C, Lundvall L, Svendsen PE, Mollerup H, Lunn TH, Simonsen I, Martinsen KR, Pulawska T, Bundgaard L, Bugge L, Hansen EG, Riber C, Gocht-Jensen P, Walker LR, Bendtsen A, Johansson G, Skovgaard N, Helto K, Poukinski A, Korshin A, Walli A, Bulut M, Carlsson PS, Rodt SA, Lundbeck LB, Rask H, Buch N, Perdawid SK, Reza J, Jensen KV, Carlsen CG, Jensen FS, Rasmussen LS, Group PT. Effect of high perioperative oxygen fraction on surgical site infection and pulmonary complications after abdominal surgery: the PROXI randomized clinical trial. *JAMA* 2009;**302**(14): 1543-1550.
18. Bickel A, Gurevits M, Vamos R, Ivry S, Eitan A. Perioperative hyperoxygenation and wound site infection following surgery for acute appendicitis: a randomized, prospective, controlled trial. *Arch Surg* 2011;**146**(4): 464-470.
19. Duggal N, Poddatoori V, Noroozkhani S, Siddik-Ahmad RI, Caughey AB. Perioperative oxygen supplementation and surgical site infection after cesarean delivery: a randomized trial. *Obstet Gynecol* 2013;**122**(1): 79-84.
20. Gardella C, Goltra LB, Laschansky E, Drolette L, Magaret A, Chadwick HS, Eschenbach D. High-concentration supplemental perioperative oxygen to reduce the incidence of postcesarean surgical site infection: a randomized controlled trial. *Obstet Gynecol* 2008;**112**(3): 545-552.
21. Mayzler O, Weksler N, Domchik S, Klein M, Mizrahi S, Gurman GM. Does supplemental perioperative oxygen administration reduce the incidence of wound infection in elective colorectal surgery? *Minerva Anesthesiol* 2005;**71**(1-2): 21-25.
22. Myles PS, Leslie K, Chan MT, Forbes A, Paech MJ, Peyton P, Silbert BS, Pascoe E. Avoidance of nitrous oxide for patients undergoing major surgery: a randomized controlled trial. *Anesthesiology* 2007;**107**(2): 221-231.
23. Stall A, Paryavi E, Gupta R, Zadnik M, Hui E, O'Toole RV. Perioperative supplemental oxygen to reduce surgical site infection after open fixation of high-risk fractures: a randomized controlled pilot trial. *J Trauma Acute Care Surg* 2013;**75**(4): 657-663.

24. Scifres CM, Leighton BL, Fogertey PJ, Macones GA, Stamilio DM. Supplemental oxygen for the prevention of postcesarean infectious morbidity: a randomized controlled trial. *Am J Obstet Gynecol* 2011;**205**(3): 267 e261-269.
25. Williams NL, Glover MM, Crisp C, Acton AL, McKenna DS. Randomized controlled trial of the effect of 30% versus 80% fraction of inspired oxygen on cesarean delivery surgical site infection. *Am J Perinatol* 2013;**30**(9): 781-786.
26. Thibon P, Borgey F, Boutreux S, Hanouz JL, Le Coutour X, Parienti JJ. Effect of perioperative oxygen supplementation on 30-day surgical site infection rate in abdominal, gynecologic, and breast surgery: the ISO2 randomized controlled trial. *Anesthesiology* 2012;**117**(3): 504-511.

VERSION 2 – REVIEW

REVIEWER	Oliveira, Ramon Universidade de São Paulo, Escola de Enfermagem
REVIEW RETURNED	27-Apr-2023

GENERAL COMMENTS	Dear authors, Thanks for the opportunity to review this manuscript. I am grateful to read a new study focusing on measures to prevent SSI, which is often a major health issue. I have raised some points to make the study sounder and provide the scientific community and healthcare providers with valuable guidance, as follows: Abstract Please check the spelling of the databases that will be employed. It would be interesting to know how the bias will be assessed. Eligibility criteria Considering the different biases that could be found in controlled studies and those in quasi-randomized investigations, why did the authors put them altogether? Identifying studies Please check the spelling of the databases that will be employed. Study selection process Will software be used to help in this phase? Data items The Centers for Disease Control and Prevention (US) claims that SSI can occur up to 30 days after surgery in which a prosthesis was not necessary and up to 90 days when a prosthesis was required. This definition is accepted by infection control scholars and healthcare providers. Taking all of this into consideration, why did the authors adopt SSI within 90 days after surgery? It will be important to know the frequency of hypothermia (or how long patients were exposed to hypothermia) besides the mean core temperature and lowest core temperature. How do the authors intend to collate the data on glucose? Blood glucose could vary intraoperatively. Please consider reporting if the sterile technique was broken during the intraoperative. Postoperative: The use of drains should be expanded for the types of drains as well.
---

	The type of dressings should be analyzed. Risk of bias: How do the authors intend to assess the bias of quasi-randomized studies, considering the use of RoB2? Another concern: Please clarify why wound classification will not be included as a subgroup.
--	--

REVIEWER	Apte, Sameer Ottawa Hospital, Surgical Oncology
REVIEW RETURNED	17-Apr-2023

GENERAL COMMENTS	Thank you for presenting this clear, and well thought out protocol for a patient level meta analysis on this important topic. I think this is a large and formidable undertaking, and I very much hope the authors are successful in obtaining this data and analysing it. I look forward to the results. I believe the authors have addressed ally concerns adequately with this revision.
--

VERSION 2 – AUTHOR RESPONSE

Reviewer 2

General response We thank the reviewer for the careful review of our manuscript and for these thoughtful suggestions. Please find any potentially relevant references at the bottom of the document.

Comment #1 Please check the spelling of the databases that will be employed.

Response We have corrected the spelling of the databases.

Abstract. Methods and analysis, page 3, lines 60-64: "Two reviewers will search medical databases and online trial registries, including MEDLINE, Embase, CENTRAL, CINAHL, ClinicalTrials.gov, and WHO regional databases, for randomised and quasi randomised controlled trials comparing the effect of intraoperative high FiO2 (0.60-1.00) to regular FiO2 (0.21-0.40) on SSI within 90 days after surgery in adult patients."

Comment #2 It would be interesting to know how the bias will be assessed.

Response Specification of the tool for assessing the risk of bias have been added to the text.

Abstract. Methods and analysis, page 3, line 68-69: "The bias will be assessed using the RoB2 and the certainty of evidence using GRADE methodology."

Comment #3 Considering the different biases that could be found in controlled studies and those in quasi-randomized investigations, why did the authors put them altogether?

Response We are aware that we are combining two different types of studies, namely controlled studies and quasi-randomized investigations. However, despite the potential biases that could be present, we have chosen to include both approaches in our research. By doing so, we aim to achieve a more comprehensive understanding of the effects of FiO₂. Additionally, incorporating multiple study designs enhances the robustness of the findings. This approach allows us to observe patterns across different studies and helps identify any discrepancies or inconsistencies that may arise. By employing methodological triangulation, we strengthen the overall validity and reliability of the research findings.

In our research, we have chosen to permit quasi-randomized controlled trials and have implemented advanced modelling techniques to account for post-randomization imbalances. This approach reduces biases that may arise from sub-optimal randomization. By optimally utilizing the available evidence, including quasi-RCT, while taking necessary modelling precautions, we aim to strengthen the overall validity and reliability of our research findings.

Comment #4 Please check the spelling of the databases that will be employed.

Response We have corrected the spelling of the databases.

Methods and analysis. Identifying studies – information sources, page 8, lines 170-171: “Medical databases will be searched, including MEDLINE, Embase, CENTRAL, CINAHL, ClinicalTrials.gov, and the WHO regional databases.”

Comment #5 Will software be used to help in this phase?

Response We have added the software tool for the title and abstract screening.

“After screening the title and abstract using Rayyan, the full text of potentially eligible papers will be retrieved and assessed.¹”

Comment #6 The Centers for Disease Control and Prevention (US) claims that SSI can occur up to 30 days after surgery in which a prosthesis was not necessary and up to 90 days when a prosthesis was required. This definition is accepted by infection control scholars and healthcare providers. Taking all of this into consideration, why did the authors adopt SSI within 90 days after surgery?

Response The longer follow-up and definition according to the author's discretion were chosen to optimize utilisation of the available data. Additionally, there is evidence suggesting that SSIs can frequently manifest after a period of 30 days. By considering this extended timeframe, the data obtained becomes more reliable and trustworthy, as it encompasses a broader scope of potential infection occurrences.2-5

Comment #7 It will be important to know the frequency of hypothermia (or how long patients were exposed to hypothermia) besides the mean core temperature and lowest core temperature.

Response We have added the suggested variables.

Table 1. Baseline and procedure characteristics, page 10: "duration hypothermia (<35°C)".

Comment #8 How do the authors intend to collate the data on glucose? Blood glucose could vary intraoperatively.

Response We agree that blood glucose levels could vary during surgery, but we intend to collect the data on glucose by collecting all available information, acknowledging that we may not receive all glucose data points intraoperatively. To capture a comprehensive view of glucose fluctuations, we plan to gather data from baseline, intraoperative, and postoperative periods. While we acknowledge that not all data may be obtained, we believe that even partial data can be valuable for analysis and interpretation.

Comment #9 Please consider reporting if the sterile technique was broken during the intraoperative.

The variables in the protocol have been carefully selected with the collaborators based on scientific interest and availability in the original trial. Although we appreciate this thoughtful suggestion and agree that it would be interesting, these data are regrettably not available in many of the studies.

Comment #10 Postoperative: The use of drains should be expanded for the types of drains as well.

The variables in the protocol have been carefully selected with the collaborators based on scientific interest and availability in the original trial. Although we appreciate this thoughtful suggestion and agree that it would be interesting, these data are regrettably not available in many of the studies.

Comment #11 The type of dressings should be analyzed.

The variables in the protocol have been carefully selected with the collaborators based on scientific interest and availability in the original trial. Although we appreciate this thoughtful suggestion and agree that it would be interesting, these data are regrettably not available in many of the studies.

Comment #12 How do the authors intend to assess the bias of quasi-randomized studies, considering the use of RoB2?

While the ROB2 tool is primarily developed for RCTs, it can accommodate quasi-randomized controlled trials by assessing bias from the randomization process. It is important to note that quasi-randomized studies may score poorly in this domain. Despite the ideal scenario of having a tailor-made tool for each study design, applying the same ROB tool to all studies enables effective comparison between different types of studies.

Comment #13 Please clarify why wound classification will not be included as a subgroup.

Subgroup analysis should be driven by a biological rationale for effect modification. The CDC wound classification represents an SSI risk classification. We have no reason to believe the intervention under investigation would work differently for the individual CDC wound classifications.

1. Rayyan---a web and mobile app for systematic reviews. Systematic Reviews 2016; 5(1): 210.
2. Holihan JL, Flores-Gonzalez JR, Mo J, Ko TC, Kao LS, Liang MK. How Long Is Long Enough to Identify a Surgical Site Infection? Surgical Infections 2017; 18(4): 419-23.
3. Bryce E, Forrester L. How Long Is Long Enough? Determining the Optimal Surgical Site Infection Surveillance Period. Infection Control & Hospital Epidemiology 2012; 33(11): 1178-9.
4. Hopkins B, Eustache J, Ganescu O, et al. At least ninety days of follow-up are required to adequately detect wound outcomes after open incisional hernia repair. Surgical Endoscopy 2022; 36(11): 8463-71.
5. Lankiewicz JD, Yokoe DS, Olsen MA, et al. Beyond 30 Days: Does Limiting the Duration of Surgical Site Infection Follow-up Limit Detection? Infection Control & Hospital Epidemiology 2012; 33(2): 202-4.

Reviewer 4

Comment #1 Thank you for presenting this clear, and well thought out protocol for a patient level meta-analysis on this important topic. I think this is a large and formidable undertaking, and I very much hope the authors are successful in obtaining this data and analysing it. I look forward to the results. I believe the authors have addressed all concerns adequately with this revision.

Response We thank the reviewer for the careful review of our manuscript and for these kind words.

VERSION 3 – REVIEW

REVIEWER	Oliveira, Ramon Universidade de São Paulo, Escola de Enfermagem
REVIEW RETURNED	25-Jul-2023
GENERAL COMMENTS	Thanks for the opportunity to review this manuscript. I am happy to read this final version of this SR protocol. Best wishes.